# Design and Parameter Identification of Wire and Arc Additively Manufactured (WAAM) Steel Bars for Use in Construction

**Johanna Müller [1],\*, Marcel Grabowski [2], Christoph Müller [3], Jonas Hensel [1],\*, Julian Unglaub [2],\*, Klaus Thiele [2], Harald Kloft [3] and Klaus Dilger [1]**

[1]   Institut für Füge- und Schweißtechnik, TU Braunschweig, Langer Kamp 8, 38106 Braunschweig, Germany
[2]   Institut für Stahlbau, TU Braunschweig, Beethovenstraße 51, 38106 Braunschweig, Germany
[3]   Institut für Tragwerksentwurf, TU Braunschweig, Pockelsstraße 4, 38106 Braunschweig, Germany
\*   Correspondence: johanna.mueller@tu-braunschweig.de (J.M.); j.hensel@tu-braunschweig.de (J.H.);
     j.unglaub@stahlbau.tu-braunschweig.de (J.U.); Tel.: +49-531-391-95517 (J.H.)

**Abstract:** Additive manufacturing (AM) in industrial applications benefits from increasing interest due to its automation potential and its flexibility in manufacturing complex structures. The construction and architecture sector sees the potential of AM especially in the free form design of steel components, such as force flow optimized nodes or bionic-inspired spaceframes. Robot-guided wire and arc additive manufacturing (WAAM) is capable of combining a high degree of automation and geometric freedom with high process efficiency. The build-up strategy (layer by layer) and the corresponding heat input influence the mechanical properties of the WAAM products. This study investigates the WAAM process by welding a bar regarding the build-up geometry, surface topography, and material properties. For tensile testing, an advanced testing procedure is applied to determine the strain fields and mechanical properties of the bars on the component and material scale.

**Keywords:** additive manufacturing; construction; WAAM; welding; steel; ESPI; design

## 1. Introduction

Advances in automation technology over the last century are the basis for additive manufacturing [1]. The idea of additive manufacturing was originally associated with rapid prototyping in order to test new design strategies in the automotive, aviation, and aerospace industry with a minimum of manufacturing effort. Today, additive manufacturing itself has become a technology that enables the fabrication of geometrically complex construction elements with efficient material usage. Additive manufacturing as a production process is also becoming increasingly interesting for tailored solutions in the construction industry. The beginnings of additive manufacturing in construction started around 10 years ago, mainly focused on extrusion techniques and concrete materials. Pioneering projects were realized by counter crafting [2], d-shape by Enrico Dini, or 3D concrete printing by Buswell et al. [3]. Since then, developments in this field have taken place at ever shorter intervals and the number of people involved in AM in the construction industry is constantly increasing [4,5].

Additive manufacturing of building components in the building industry is not only limited to processes for cement-bound components. The idea of creating metal components by layers of welded seams was already being considered in the 1930s [6]. However, additive manufacturing from metals in construction has only recently become the focus of attention and is the subject of research in the construction industry [7,8]. Arup [9] and Joosten with MX3D [10] demonstrated how steel components can be manufactured by additive manufacturing. Camacho et al., however, showed that the main

field of application of metallic AM components currently lies in aircraft construction, automotive engineering, and health technology [11].

The motivation for additive manufacturing in the construction industry results among other things from the low labor efficiency in comparison to automated machines as well as from aspects, such as occupational safety, accident rate, and low construction quality due to inadequate training of employees [12,13]. In addition to these points, increasing material efficiency, minimizing waste, reducing building mass, and, consequently, reducing energy consumption in the construction of the component or building are reasons for additive manufacturing in construction.

The production of components is usually carried out in subsequent steps, which is why the construction industry and the production process can be considered as being very fragmented. For this reason, additive manufacturing should not be regarded as a replacement for previous methods of construction, but as an alternative manufacturing process for components that can only be manufactured by traditional methods at great expense. The manufactured geometries can increase in their complexity without increasing costs and effort in the same measure. The aim is to reduce the sum of all individual parts of a building in order to simplify the construction of the building. By digitalizing the whole design process, the number of necessary steps can be reduced compared to traditional design processes. Design, optimization, and preparation for the additive manufacturing of components is done digitally. Work steps, such as transformation of the three-dimensional digital model into two dimensions for worksite planning, are not necessary. From the planning to the finished component, all information is available digitally and allows the creation of a digital twin, which can be transferred into a building information modelling system (BIM).

For architecture, additive manufacturing changes the limitations of fabrication. It opens up the possibility of adapting the design down to the smallest detail without being subject to the restrictions of today's common manufacturing methods. Particularly in the field of steel construction, the design based on manufacturing from semi-finished products, such as plates and prefabricated profiles. Therefore, there is a high potential of geometric freedom to be achieved by AM in this area.

Processes for the additive manufacturing of steel structures can be subdivided into powder-based (selective laser melting (SLM), laser metal deposition (LMD), electron beam melting (EBM)) or wire-based (WAAM, electron beam freeform fabrication (EBF³), laser-engineered net-shaping (LENS)) processes [14,15]. In WAAM processes, the wire-shaped additive materials are either supplied to the energy source (gas tungsten arc welding (GTAW), plasma and laser) and are molten, or the wire itself is the electrode, like in gas metal arc welding (GMAW) processes [16].

The advantages of wire-based technologies in comparison to powder-based processes are the higher deposition rate in combination with a better material utilization and lower material cost, making it suitable for applications in the construction sector. Powder-based technologies are more suitable for complex geometries with high requirements on the geometric precision [17]. Further, the component size in the SLM process is limited regarding the component size, since the dimensions of the component cannot exceed the powder bed size. The WAAM process utilizes CNC or robot kinematics and the AM element is, hence, theoretically not limited in size.

WAAM as a field of research gains more and more attention. Numerous investigations focusing on the application of WAAM in lightweight constructions using titanium or aluminum alloys as filler material [18–20]. The driving forces are the automobile and aerospace industry. Other works focusing on steel as construction material, where the fields of application are pipe couplings and flanges, special machines, and plant construction, mainly driven by the naval and crane industry [17]. In the construction sector, studies on the WAAM with steel electrodes of nodal joints and beam reinforcements are available. Their focus is mainly on the constructive design potential or on the generation process itself [7,21]. Other works identified the processing capabilities of WAAM in general by investigating several materials, power sources, and welding processes, such as GMAW, GTAW, and plasma welding. Studies provide guidelines for the additive process selection, aiming at improved material properties. Advanced process modifications, such as workpiece and wire oscillation, modulated power supplies,

and use of shielding gas or different cooling strategies, were applied [22–24]. Furthermore, there are publications which investigate the thermal cycles and the reheating of prior layers during the WAAM of wall-like structures using high-strength low-alloy (HSLA) steel [25] and duplex stainless steel [26]. According to Rodrigues et al., during reheating, recrystallization may occur, leading to fine-grained microstructure or coarse grains in the case of high peak temperatures. However, the microstructure and thus the mechanical properties are affected by the thermal cycles and the amount of layers being reheated during the welding of subsequent layers.

WAAM makes it possible to increase the geometric complexity of a component without increased manufacturing effort. At present, there is no unified methodology for the design of component geometry [27]. The most common approaches result either from the manufacturing process or from the optimum use of the material. The first approach is a bottom up process and is based on the path planning of the component to be manufactured [28]. The second approach is a top down process and is usually based on topology optimization. Questions about the aesthetic qualities of the component have not been discussed yet [29]. However, it is of essential importance for architecture and the building industry.

Each working step, from the creation of the basic geometry, through topology optimization and adaptation for the manufacturing process, to the verification of the load-bearing capacity and stability, creates its own geometry. Restrictions should already be defined during the design of the initial geometry in order to meet the architectural requirements of the overall structure. Here, traditional design concepts should be abandoned and possibilities of additive production should be decisive. Good design only appears possible if there is a precise knowledge of the effects of each individual design step. One of these design steps must consider and include the process parameters and boundary conditions for the manufacturing of a component by WAAM. Structures to be produced by WAAM can be subdivided into various substructures, such as bars, surface elements, and volume elements.

In this article, the focus is on steel bars not only as a substructure, but also as an independent structural component. The possibility of production and the appearance of surface topography as well as homogenous material properties are of special interest. Problems, such as sufficient space for the welding torch, inclination of bars, number of crossing points, and resulting build-up, and path planning strategies must be taken into account at the design stage.

In addition to using bars as a substructure in a structural component, WAAM manufactured bars can also be used as reinforcement in reinforced concrete. Figure 1 shows reinforcement arrangements for additively manufactured concrete components. The alignment of the reinforcement is still strongly oriented to the geometry of traditional prefabricated reinforcement elements. WAAM allows a more efficient arrangement of the reinforcement and extends the use of additive manufacturing in concrete construction. Due to the higher degree of freedom of WAAM manufactured bars, reinforcement can be arranged according to the tensile stress curve.

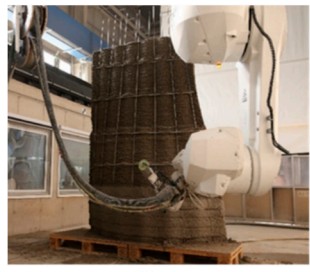 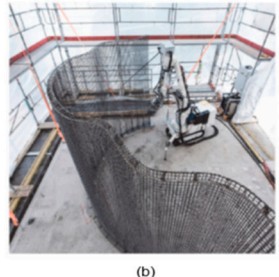 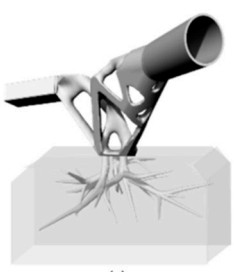

（a）　　　　　　　　　　　（b）　　　　　　　　　　　（c）

**Figure 1.** (**a**) Construction of a reinforced concrete wall in the digital building fabrication laboratory (DBFL) at the Institute for Structural Design (ITE). (**b**) Curved mesh of conventional reinforcement bars constructed by an in situ fabricator [30], reproduced with permission from Norman Hack, Ph.D. Thesis: Mesh Mould: A Robotically Fabricated Structural Stay-in-Place Formwork System; published by ETH Zurich, 2018. (**c**) Root-like structure for anchoring steel components in concrete.

An extended application for WAAM-manufactured reinforcement is the anchoring of steel components to reinforced concrete. Figure 1c schematically shows the possible root-like anchoring of a steel construction node to a concrete foundation. The use of bars as reinforcement requires different surface properties than the nodal joints produced by WAAM. The curvature of the surface is decisive for the bond between the concrete and reinforcement. Control of the surface topology would be advantageous, especially with regard to root-like anchoring, as shown in Figure 1c.

In order to use components or production types in the European Union, a European Technical Assessment (ETA) is required. The ETA assesses the performance of a construction product with respect to its essential characteristics. ETA provides a way to CE-certification (European Conformity certification) for construction products that are not or are not fully covered by a harmonized standard. This certification process can only be applied to WAAM components with limitations. The evaluation of process and material irregularities and flaws needs to be addressed in future certification concepts for WAAM in construction. Therefore, existing certification concepts need to be extended. A possible solution is the concept of a digital twin, which holds a set of any virtual information that describes the actual physical manufactured product on different scales [31]. This model is already being used in aircraft production for general components as well as for additive manufactured parts. Currently, this concept will be extended by blockchain technology to ensure the traceability of the process chain steps [32]. It is necessary to adapt this concept to the particularities of the construction industry. The main challenge for a successful AM technology adoption in the construction sector is the certification of AM components for safe use in the buildings of infrastructure. For this purpose, basic knowledge on the relationship between the manufacturing process and material properties of WAAM components first must be determined.

The determination of the material properties, more specifically, the mechanical properties of additively manufactured bars, differs from conventional material testing due to features, such as topography (notches). Further, different manufacturing irregularities, in terms of WAAM weld irregularities and flaws, need to be addressed by the tests and evaluated. For mechanical testing of AM-parts, like walls or cylinders, the well-established way is to section samples in the form of a standard tension test specimen from the AM-structure, which are prepared and tested according to any ISO or ASTM standard [33,34].

Root-like anchoring elements, cast in concrete or reinforcement elements, where an uneven topography or surface irregularities are desired, may show losses in mechanical properties due to notches. Those surface irregularities can be sources for premature failure. So, for applications with cyclic loading, an even surface is more favorable and the bars should be post-processed, i.e., machined. However, those reinforcement elements are not meant to be post-processed, so the influence of the topography on the elastic behavior is essential information when investigating the mechanical properties. So, cutting out standard conform specimen geometries is not an option because the topography information would be lost.

The new approach for the mechanical testing of WAAM-bars, which is introduced here, is to take a standard round tensile test specimen [35] and replace the cylindrical middle part with an additively manufactured bar.

The aim of this work is to investigate the influence of welding process parameters on the surface topography of steel bars and the local strain distribution as a consequence of mechanical loading. Furthermore, the energy inputs and cooling rates of different welding processes are related with the resulting hardness, microstructure, and the mechanical properties.

## 2. Materials and Methods

The manufacturing process of the steel bars was carried out on a 10 mm thick S355N (1.0545) substrate plate. The welding material itself was a solid wire electrode of unalloyed steel G4Si1 (1.5130) with a diameter of 1.0 mm. The chemical composition of the materials is given in Table 1.

**Table 1.** Chemical composition of the used materials.

| Material | Chemical Composition (wt.-%) | | | | | | | | |
|---|---|---|---|---|---|---|---|---|---|
| | C | Si | Mn | P | S | Cr | Mo | Ni | Cu |
| G4Si1 | 0.061 | 0.845 | 1.66 | 0.025 | 0.0139 | 0.035 | 0.006 | 0.044 | 0.088 |
| S355N [1] | 0.2 | 0.5 | 0.9–1.65 | 0.03 | 0.025 | 0.3 | 0.1 | 0.5 | 0.55 |

[1] Nominal composition.

The experimental welding setup was composed of a Fronius Cold Metal Transfer (CMT) Advanced 4000 R power source (Fronius International, Pettenbach, Austria), a handling robot, temperature gauges, and process periphery, as shown in Figure 2. The torch as well as the pyrometer were mounted on the KUKA KR22 robotic handling system (KUKA AG, Augsburg, Germany). The pyrometer was an optris CTlaser 3MH2 with a measuring range from 200 to 1500 °C. The substrate plate was attached to the positioning table.

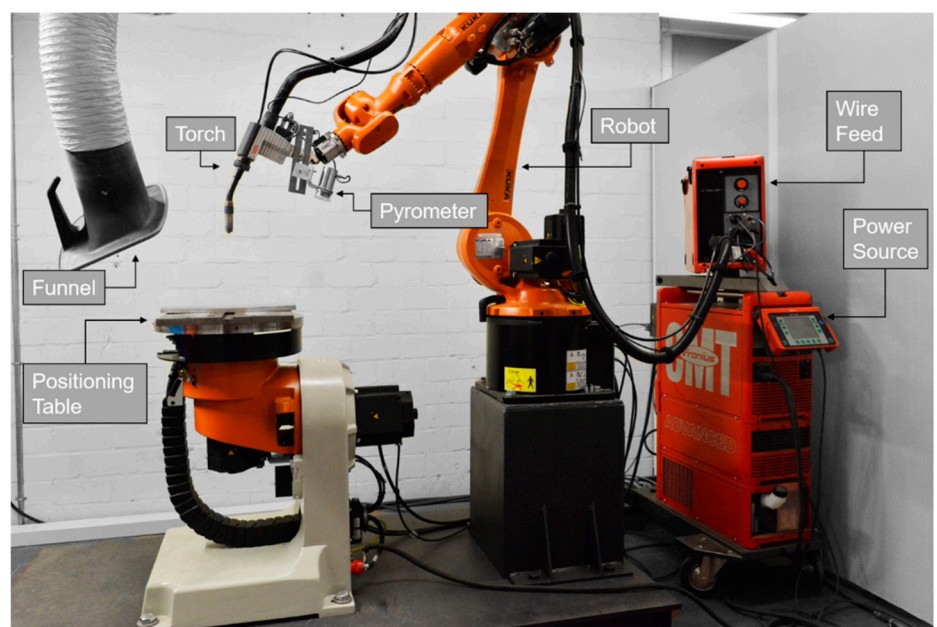

**Figure 2.** Experimental setup for the WAAM of steel bars [34].

One special feature of this power source is the reversing wire electrode controlling short circuit phases and droplet transfer to the weld pool. That leads to a minimal-spatter ignition due to the almost current-free material transition. The current can be kept very low for a longer time, thus significantly reducing the heat input to the piece [36,37]. This is what makes such modified short arc processes so suitable for WAAM applications.

In this investigation, two different CMT process variants were used. The first was a CMT standard process and the second was the CMT cycle step, which reaches even higher short-circuit percentages than the CMT standard by controlled switch-offs of the arc, leading to a lower heat input [37]. The cycle step also convinces with a precise control, which allows the choice of the exact number of droplet transitions per cycle instead of the welding time, like in CMT standard processes [38]. In addition to the CMT processes, one steel bar was manufactured with an elmatech DV36 L(W) GMAW (denoted in the following as conventional GMAW) power source for comparison. The welding parameters for each process are depicted in Table 2. As shielding gas M21 (82% argon and 18% $CO_2$) with a constant flow of 12 L/min was used.

**Table 2.** Welding parameters.

| Parameter | Unit | Conventional GMAW | CMT Standard | CMT Cycle Step |
|---|---|---|---|---|
| Wire Feed | m/min | 10.6 | 5 | 9.4 |
| Welding time | s | 1.5 | 2.4 | 1 |
| Material transitions | Droplets/cycle | - | - | 100 |
| Current | A | 218 | 158 | 204 |
| Voltage | V | 27.6 | 11.1 | 16.4 |
| Energy per layer | kJ | 9 | 4.21 | 3.35 |

As shown in Figure 3a, the bars were welded point by point, while the torch was moved up in the build-up direction. According to the deployed process, one point was welded either for a certain time (conventional GMAW, CMT standard) or for a certain number of droplets (CMT cycle step). Subsequently, the torch was automatically moved up for 2, respectively, 1.6 mm (depending on the resulting layer height) and a waiting period followed, until a constant interpass temperature of 200 °C was reached. Then a new cycle began. The $t_{8/5}$-times in the middle layers averaged for both CMT processes was 16 s.

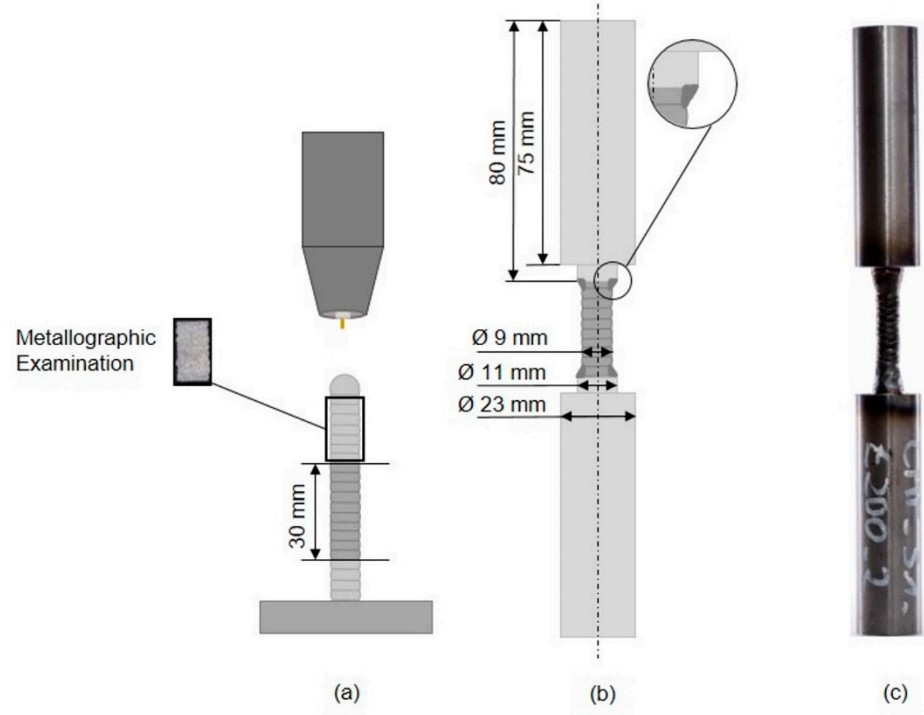

**Figure 3.** Schematic of the specimen preparation: (**a**) Welding the bar, (**b**) specimen preparation for the tensile tests, and (**c**) ready-to-test specimen.

For characterization of the bars regarding their mechanical properties, the bars had to be in a proper shape for execution of the tensile tests. Especially concerning the clamping cylinder in the tensile test machine, the specimens had to be an even shape to not initiate bending moments into the bar itself or into the machine.

The most obvious specimen shape is a round tensile test specimen in dependence on DIN 50125. Therefore, a piece with a length of 30 mm was cut out of the bar and then welded between two cylindrical ends of standard round tensile test specimens, which were turned to a diameter of 11 mm at one side for better accessibility during the welding of the fillet, as shown in Figure 3b. The joint was welded with a TIG-process using 110 A and no filler material. For all samples, computer tomography scans with a "GE v|tome|×240s Research with micro focus tube" were made. Subsequently, the tensile tests followed. Additionally, for all three samples, the surface was characterized using a laser scanner

(optoNCDT 1800, micro-optronic, Langenbrück, Germany), micrographs were prepared, and hardness tests were executed.

The experimental setup for the tensile testing is shown in Figure 4. The specimens were mounted on a tensile testing machine (MTS 810 Servo Hydraulic Testing System, MTS Systems GmbH, Berlin, Germany) and two optical measurement systems were used for the tensile tests. In order to take sufficient account of surface influences and the anisotropic inhomogeneous material behavior, full field strain measurements were taken with the Electronic Speckle Pattern Interferometry (ESPI; Q-300, Dantec Dynamics GmbH, Ulm, Germany). Because the ESPI requires a high positional accuracy relative to the specimen, it is mounted on a 3D-positioning system, which allows the positioning with an accuracy of 1 μm. Additionally, a laser extensometer (P-50, Fiedler Optoelektronik GmbH, Lützen, Germany) was used to verify the measured strain maps during the test. This second measurement was done over the whole length of the welded part of the bars.

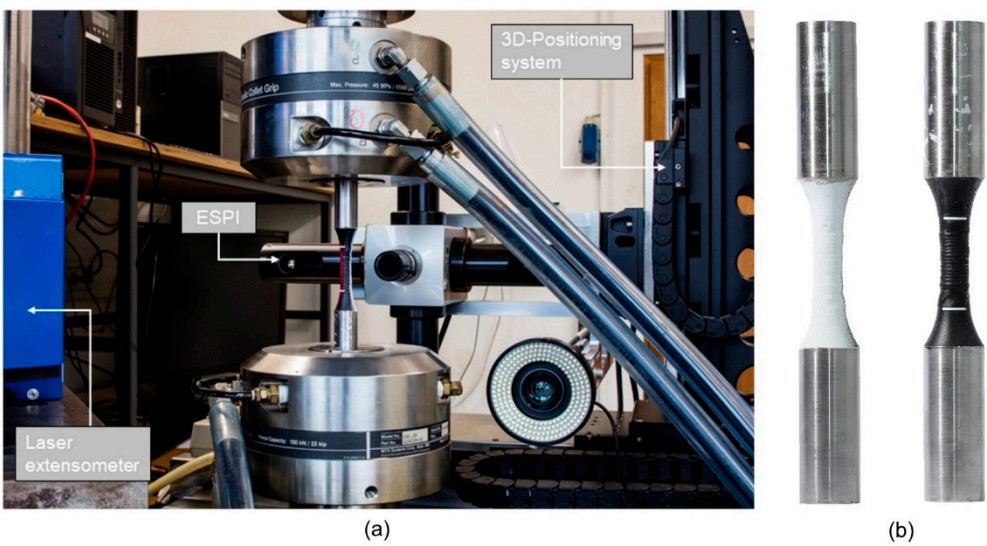

**Figure 4.** (**a**) Experimental setup; (**b**) specimen sprayed with lime powder (left) and black lacquer (right).

ESPI employs a coherent laser that illuminates the specimen from different positions and monitors the change in the intensity of the produced interference due to the displacements with a CCD camera. The setup used here can measure displacements of the surface in all three spatial directions. The measured displacement fields are then numerically differentiated in order to obtain strain maps. One advantage of the ESPI for the measurements on the material scale is its high displacement measurement resolution (up to $10^{-6}$ m) [39]. Due to this high resolution, ESPI is often used for measurements of displacement and strain due to thermal loading on small parts in the sectors of aerospace technology or electrical engineering. ESPI measurements on aluminide layers on a MAR 247 nickel alloy were also shown to be able to identify localized areas of deformation concentration as a starting point of damage under cyclic loading [40].

The use of standard tensile test specimens for the determination of mechanical parameters was not directly feasible here. Conducting tensile tests until failure with the topography "as-welded" would not allow the calculation of definitive material parameters, except the component's bearing load, as the cross section is not well-defined. Turning the whole specimen leads to a loss in information as the surface would be removed to create a defined measuring section. Therefore, an advanced testing strategy was applied here, which was divided in two stages of tensile tests with an additional metallographic examination parallel to the tensile tests, as shown in Figure 5.

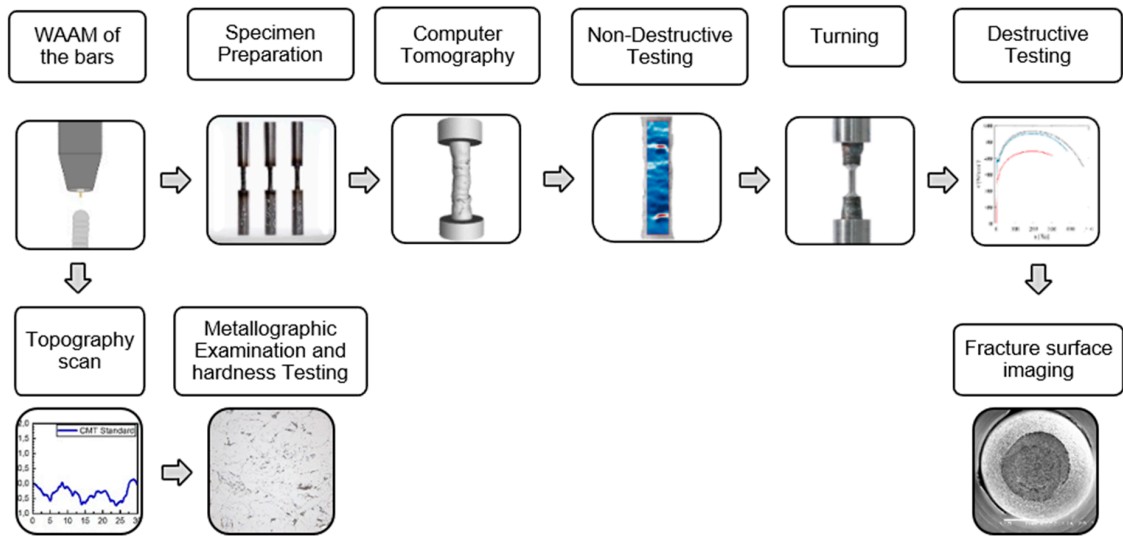

**Figure 5.** Advanced testing strategy for the identification of strain fields and material parameters.

The non-destructive first stage of testing was applied to the sample with the topography "as-welded" as shown in Figure 5. The aim of the first tests was to analyze the influence of the weld process-dependent surface topography and the inhomogeneous material behavior in the elastic state. The load had to be limited in order to avoid plastic deformations and was identified in preliminary tests to a maximum force of 5 kN. Due to the high sensitivity of the ESPI, the measurements had to be conducted after load step increments of 0.5 kN with holding times of 5 s each. Afterwards, the measured deformations and strains of each load step were summed up. Simultaneously to the ESPI measurement, the laser extensometer measured the strains over the whole length of the welded specimen (30 mm) to verify the strains measured with the ESPI.

Both optical measurements required a different surface preparation. For the ESPI measurement, an optically rough and reflective surface is required. This was achieved by spraying on a layer of lime powder. On the other side of the specimen, a primer with a matt black lacquer was applied to serve as a non-reflective background that could be distinguished from the white measuring marks, which is displayed in Figure 4b.

In order to obtain the material parameters, a second stage of destructive tensile tests was carried out. For these tests, a well-defined transversal section was needed to calculate the resulting stress and to derive the stress–strain diagram with all needed design parameters. For this reason, the middle part of the bars was turned to an even transversal section with a length of 10 mm. To ensure that the failure of the specimen occurred in the desired section and not in the fillet weld, a diameter of 4 mm was chosen for the transversal section. The strains during the test were measured with the laser extensometer. For this, the measurement markers were placed on the outer edge of the transversal section at a distance of ca. 10 mm.

After the destructive testing, the fracture surface was examined using a scanning electron microscope (JEOL JSM 6480, Japan Electron Optics Laboratory, Akishima, Japan) to obtain information about the fracture mechanism.

## 3. Results

### 3.1. Comparison of the Build-Up Height and the Resulting Diameters

The resulting bar diameters, including minimum and maximum values, as well as the mean build-up heights of the single layers were examined and are displayed in Figure 6. Furthermore, the

energy input per layer was calculated by Equation (1), since there is no welding velocity for calculating the regular energy per unit length:

$$E = U \times I \times t_{weld} \tag{1}$$

For measurement of the welding time per layer ($t_{weld}$), the current ($I$), and the voltage ($U$), a measurement device (HKS weld monitoring system) was connected to the power source. The according energy per layer is also shown in Figure 6.

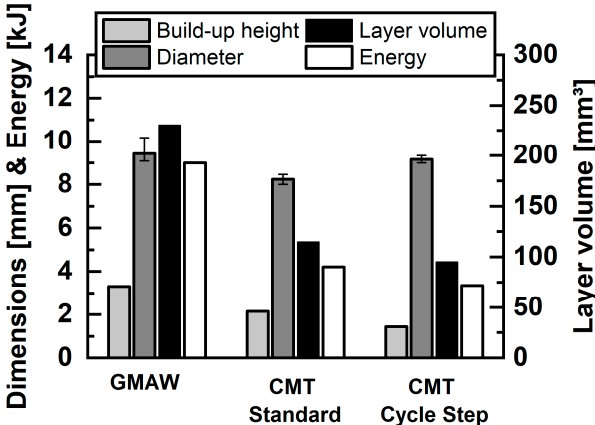

**Figure 6.** Layer build-up height, diameter, layer volume, and the corresponding energy input of different welding processes.

The conventional GMAW obtained the highest build-up rate with more than a 3.3 mm layer height as well as the thickest bar diameter with almost 10 mm. The CMT cycle step reached 1.4 mm as the lowest layer height and a bar diameter of 9.16 mm. The CMT standard process achieved layer heights of 2.15 mm and diameters of 8.2 mm. Since the build-up height correlates with the energy per layer while the diameter does not, the layer volume was also calculated and is depicted in Figure 6. The layer volume increased with higher energy input per layer.

### 3.2. Surface Topography

Next to the build-up volumes and diameters of the bars, the topographical properties of the WAAM bars were specified. It can be seen from the results in Figure 7 that the conventional GMAW led to a non-uniform surface with a minimum to maximum range of 1.4 mm, determined along a measuring section of 30 mm. The surface topography of the CMT standard specimen was smoother and the layers were built more regularly. The waviness was in the order of 0.89 mm. In comparison, the CMT cycle step process led to the most uniform and even surface. The range between the minimum and maximum was 0.35 mm.

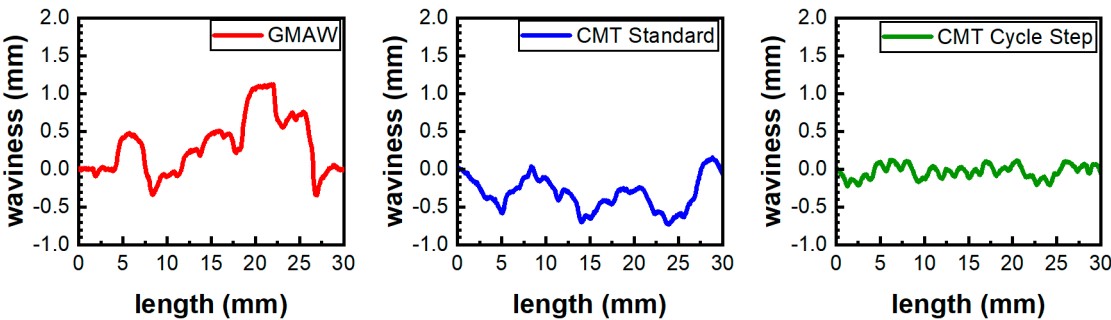

**Figure 7.** Surface laser scans along the build-up direction.

### 3.3. Microstructure and Hardness

Subsequently, deposited layers reheat weld metal during the manufacturing process. The reheating causes a change of the microstructure. Especially, the refined geometry of the bars manufactured here lead to heat accumulation and comparably long $t_{8/5}$-times in combination with high peak temperatures. Micrographs showing the weld layer geometry and the resulting hardness are given in Figure 8.

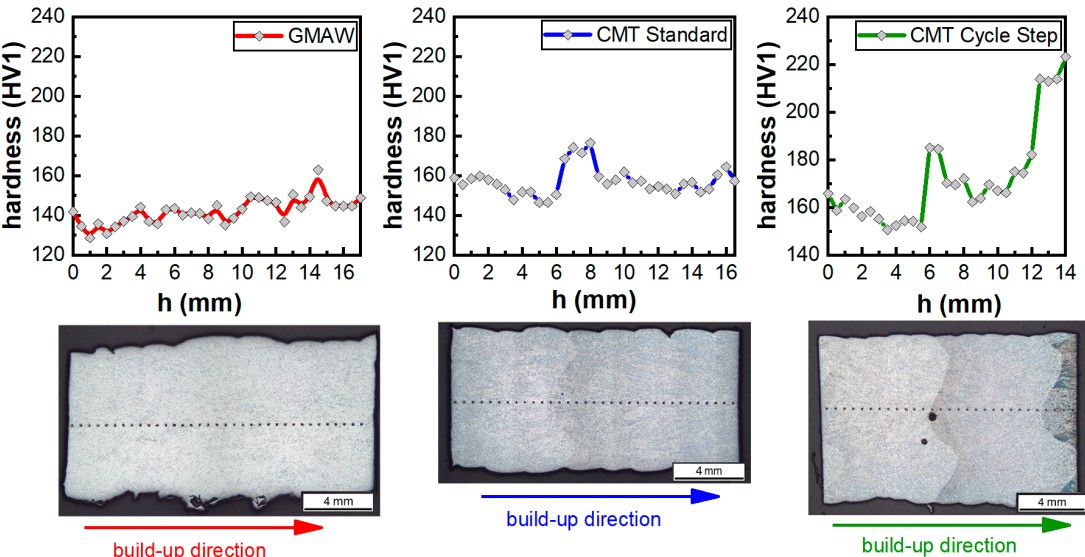

**Figure 8.** Hardness distribution in the build-up direction.

The averaged hardness increased with decreasing energy input. In the conventional GMAW process (9 kJ per layer), the hardness ranged from 128 to 163 HV1; in the standard CMT process (4.2 kJ per layer), the hardness varied between 146 and 176 HV1; and in the process with the lowest energy input per weld point (3.35 kJ), the hardness increased from 150 HV1 in the lower layers to 223 HV1 in the higher layers. In the pictures on the bottom of Figure 8, one can see that due to the high heat input in the conventional GMAW weld, the layers melted together and an even homogenous secondary microstructure developed. Hence, the hardness did not vary as much as in the CMT cycle step specimen. Looking at the two CMT processes, one characteristic is the hardness increase at the transition zone between two layers. Right when the next layer started, the hardness rose significantly and decreased subsequently.

Figure 9a shows the microstructure of the conventional GMAW specimen consisting of mostly ferrite and small fractions of bainite and perlite. Here, the grains were comparatively large. The grain growth is most likely a result of the peak temperatures well above the austenitization temperature, $A_3$, and the comparably long holding times.

In general, the microstructure of the CMT processes in the middle and on the top of each layer is a fine-grained bainite-ferrite structure. In the micrographs in Figure 9b,c, the grains appear smaller in the area of high hardness and a higher amount of bainite can be detected. Figure 9c shows at the right side the top layer of the bar, which was welded last. The resulting primary microstructure not affected by subsequent weld layers is composed of ferrite and bainite fractions in an acicular structure. However, the secondary microstructure resulting from both CMT processes did not vary significantly while the high energy input of conventional GMA welding led to a significantly lower, but also more homogeneous, hardness. This is expected to result in a variation of the tensile properties compared with Section 3.6.

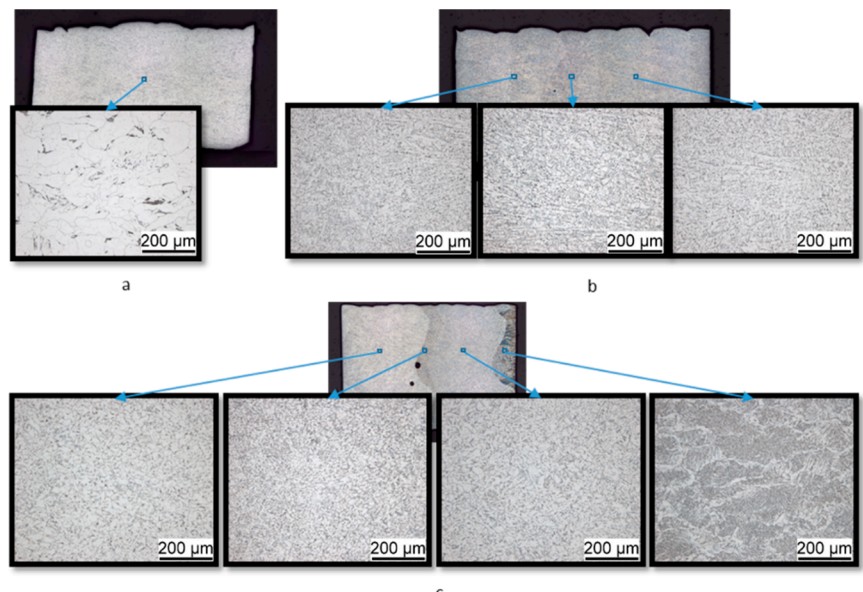

**Figure 9.** Micrographs of the specimen manufactured with (**a**) conventional GMAW, (**b**) CMT standard, and (**c**) CMT cycle step.

General effects of cooling rates on microstructure and hardness are given in welding time-temperature-transformation (TTT) diagrams. Figure 10 shows the transformation behavior of a G4Si1 at rapid cooling. However, the chemical composition of this welding wire varies from the one used here, but general effects can be explained. Fast cooling rates lead to more bainite and less ferrite, resulting in higher hardness. Low cooling rates result in low hardness and mainly a ferrite/pearlite microstructure. Accordingly, the hardness of deposited G4Si1 can vary depending largely on the welding parameters.

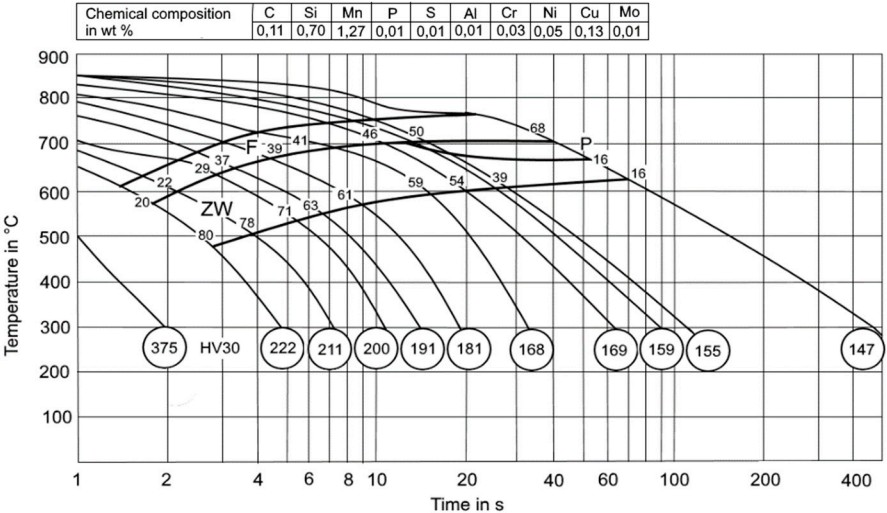

**Figure 10.** Welding time-temperature-transformation (TTT) diagram for G4Si1 and peak temperatures of 1350 °C, according to [41].

The fine-grained microstructure is an indicator for low heat input while welding subsequent layers and moderate peak temperatures above $A_3$. The slightly harder regions at the layer interface were reheated less during subsequent thermal cycles. Thus, annealing effects are less prominent. The result is a heterogeneous microstructure in the build-up direction with varying mechanical properties.

### 3.4. Computer Tomography

The characterization of the three specimens in terms of the porosity and cavities was carried out with computer tomography before the tension testing. The results of the porosity analysis of the CT scans are displayed in Figure 11. On the left side, there is the conventional GMAW bar with a comparatively low amount of pores and porosity. The diameter of the pores ranged from 100 to approximately 1000 μm. The pores in the bar welded by the CMT standard process were both small and rare. The highest detected pore diameter measured between 100 and 200 μm. While the size of the pores in the CMT cycle step bar are comparable to the CMT standard bar, the amount of pores or cavities exceeds that of the CMT standard bar many times over. The differences in terms of porosity are a result of the weld pool size and the varying degassing behavior during welding. Larger weld pools in combination with short welding times (conventional GMAW) cause solidification pores of a distinct size. Smaller weld pools (CMT cycle step) and longer welding times (CMT standard) are favorable for low pore sizes. However, short welding times in combination with small weld pools lead to a high number of small pores.

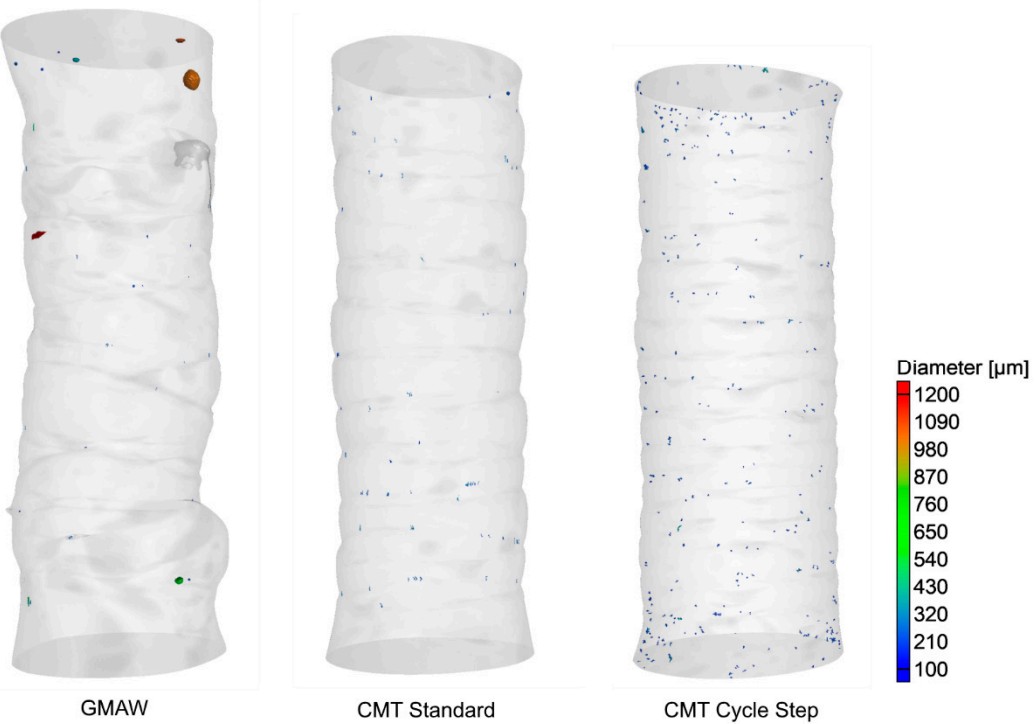

**Figure 11.** Computer tomography scans with porosity analysis of the conventional GMAW bar, the CMT standard bar, and the CMT cycle step bar.

### 3.5. Full Field Strain Measurements

The material behavior and the influence of the surface topography were examined by full field strain measurements using ESPI during uniaxial tensile tests in the elastic region. To compare the three specimens, the measured strain maps (longitudinal strains) were related to the global longitudinal strain, $\varepsilon_g$, simultaneously measured with the laser extensometer on each specimen. Figure 12 shows a surface photograph with the ESPI measurement area and the related strain map of each specimen at a load of 5 kN.

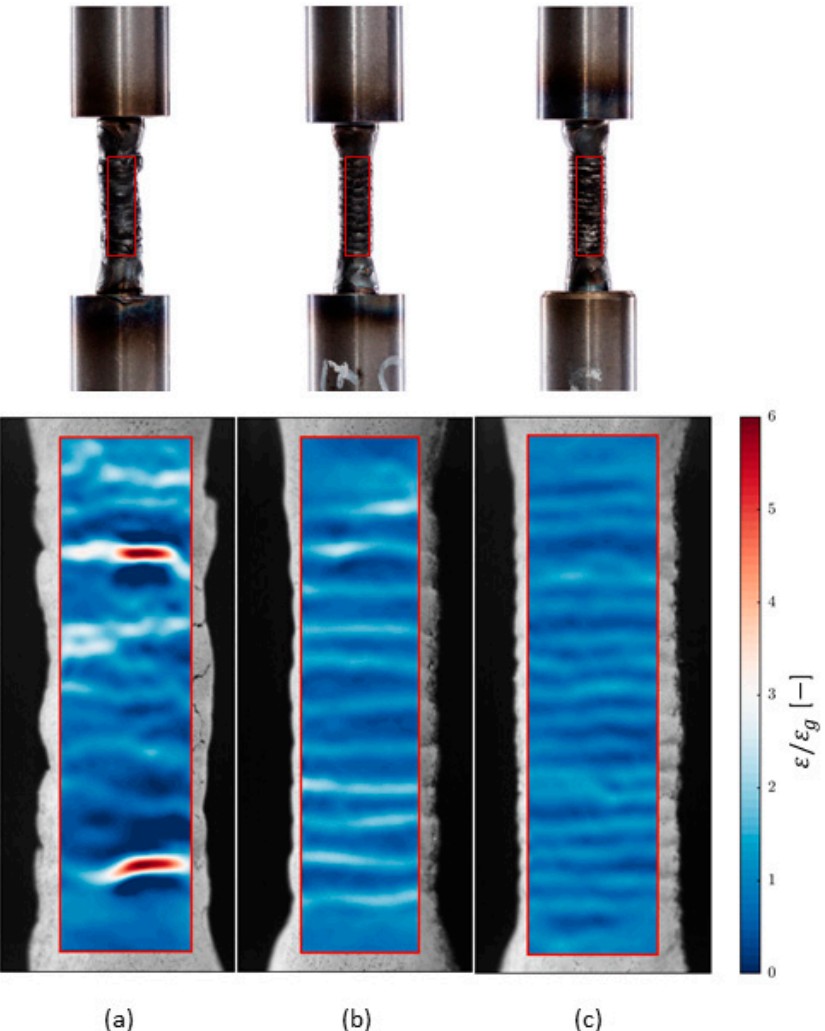

**Figure 12.** Photograph with the measurement area and local strain maps (longitudinal strain) related to the global strains of (**a**) conventional GMAW, (**b**) CMT standard, and (**c**) CMT cycle step under tensile load.

The ratio from local to global strain ($\varepsilon/\varepsilon_g$) allows detection of areas with different material parameters or strong variations in the surface. The local strains in the conventional GMAW bar showed the largest deviations from the measured global strains with a ratio of up to 6 in two distinct areas of the specimen, as displayed in Figure 12a. The CMT standard bar showed a more even distribution with ratios of up to 2.5. In comparison, the CMT cycle step bar showed the most even strain distribution with a maximum ratio of 1.5. While the layered structure was clearly visible in both CMT bars, the conventional GMAW bar showed two concentrations, which can be attributed to strong deviations in the surface topography. Effects of the microstructure or heterogeneous hardness on local strain distribution were not detectable at this load level.

Due to the high local strain concentrations, especially in the conventional GMAW bar, local plastic deformations might have occurred. In the event of a further load increase, the conventional GMAW bar would most likely fail at one of the localizations. This shows that a uniaxial tension test until failure with this topography would not provide representative results about the mechanical properties, except the ultimate bearing load. Therefore, the advanced testing strategy applied here seems to represent a better overall solution. A rated value similar to the strain ratio presented here could later be used as a notch factor corresponding to the different bars. Therefore, these measurement results can also be used to examine the material behavior of WAAM bars in regards to cyclic loading.

*3.6. Mechanical Properties*

The second stage of testing consisted of uniaxial tension tests until failure with a further machined specimen geometry, comparable to conventional tensile testing. For each type of bar, one specimen was tested until failure. A laser extensometer was used to measure the integral strains during the test on the length of the transversal section (10 mm). Figure 13 shows the obtained engineering stress–strain curves for all three specimens. The resulting stress was calculated based on the 4 mm diameter in the predefined measurement section.

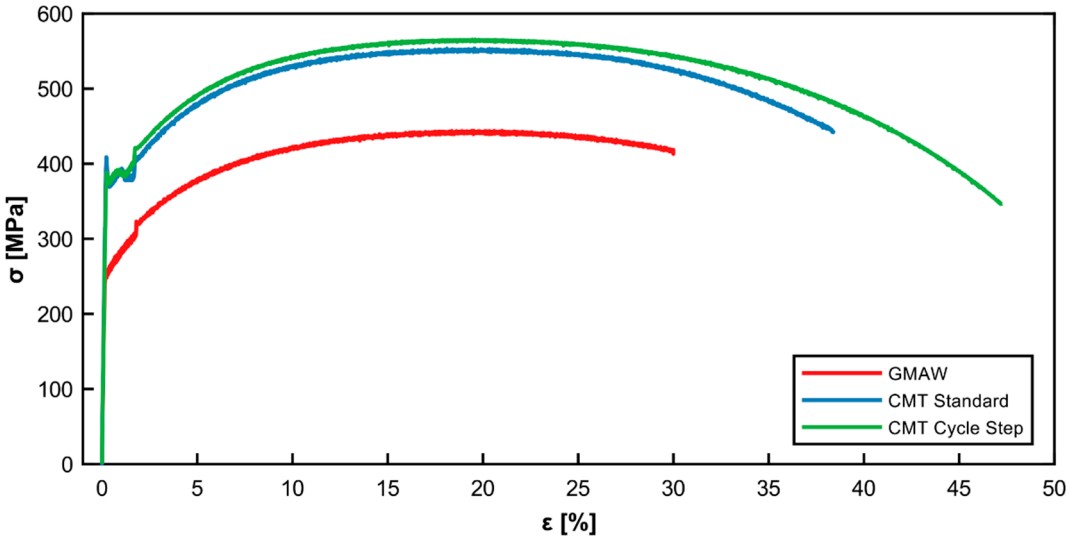

**Figure 13.** Stress–strain diagram for all three specimens.

The stress–strain curves show a very ductile material behavior with large ultimate elongations, especially for the CMT bars. A clear yield point was also recognizable for all three specimens as is expected from the used unalloyed weld metal. All mechanical properties, as listed in Table 3, were calculated in accordance to DIN EN ISO standards [42]. Table 3 also includes the corresponding properties of the weld metal data sheet from the welding wire used.

**Table 3.** Mechanical properties of the three bars and the welding wire.

| Properties | Unit | Conventional GMAW | CMT Standard | CMT Cycle Step | Weld metal (Data Sheet) |
|---|---|---|---|---|---|
| Modulus of Elasticity $E$ | GPa | 180 | 198 | 191 | - |
| Yield Stress $R_{eH}$ | MPa | 265 | 410 | 390 | ≥460 |
| Ultimate Tensile Stress $R_m$ | MPa | 445 | 554 | 565 | 530–680 |
| Ultimate Elongation A | % | 49.7 [1] | 46.3 [1] | 4.7 [1] | ≥20 |

[1] Due to the geometry of the specimen, the elongation at break is not comparable to standard tensile test specimens [35].

All the bars showed a linear elastic material behavior with a modulus of elasticity ranging from 180 to 198 GPa. The yield stress, $R_{eH}$, for all three bars and the ultimate tensile stress, $R_m$, for the conventional GMAW bar is lower than nominal values from the corresponding weld metal data sheet. For both CMT bars, the $R_m$ lies in the desired range. In comparison, the conventional GMAW bar shows the lowest values in terms of strength properties. The results are in agreement with findings from metallographic investigations and hardness tests.

*3.7. Fracture Surface Imaging*

After destructive testing of the bars, the fracture surfaces were examined by help of scanning electroscope micrographs. Figure 14 shows the fracture surface of each bar in two different magnifications. The macroscopic images of all the specimens depict significant area reductions resulting from plasticity. This effect is most distinctive at the specimen welded by conventional GMAW. However, all fracture surfaces show dimples at high magnification. On the surface of the conventional GMAW specimen, there are also artefacts, which may be silicate segregations. In comparison to the GMAW specimen, the CMT specimens show fracture surfaces with indications for multiple crack initiation sites. A rougher surface indicates more ductile fracture behavior.

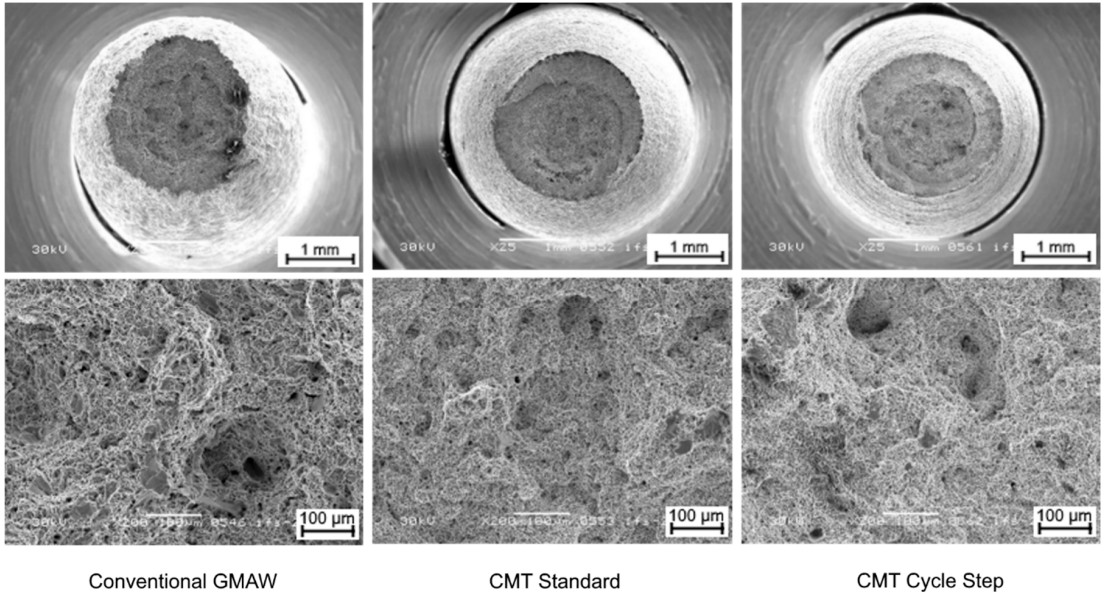

**Figure 14.** Fracture surface images of the bars after destructive tensile testing.

## 4. Discussion

Steel bars manufactured by the WAAM process demonstrate the use of AM in construction. For this study, several samples were manufactured with each process to identify the influence of welding parameters. However, only one sample of each process was tested until fracture. The overall objective was the identification of an advanced test routine, which can be applied to further WAAM components as well. The steel bars, as manufactured and tested here, could represent reinforcement elements in concrete applications or standalone, respectively, parts of complex larger steel components structural members. Further, it is demonstrated that the use of manufacturing parameters, such as the welding process and the energy input, may result in a severe influence on the component and material properties. The energy reduced short arc welding processes led to a heterogeneous microstructure with hardness peaks at the layer interfaces, probably resulting from low annealing temperatures between $A_1$ and $A_3$. This is usually not favorable as mechanical properties become heterogeneous as well. Higher interpass temperatures could solve this issue also for welding processes with low heat input. However, the resulting lower weld pool viscosity may affect the resulting surface topography as well. The results from the microstructure and hardness testing of WAAM bars as well as the surface fracture images correspond well with theoretical findings from the literature (e.g., TTT-diagrams), weld parameters, and results from tensile tests. The conventional GMAW bar with lower, but homogeneous, hardness showed lower ultimate strength and higher areas of reduction at the fracture compared to the CMT-welded bars. However, the peak temperatures and annealing times should be controlled more intensively in future research as they control the grain size.

The surface topography and the geometric dimensions, here the bar diameter, depend on the process characteristics. A homogeneous diameter is preferable in terms of mechanical material utilization. However, in concrete applications, rougher surfaces may be beneficial due to better load transfer from concrete to reinforcement bars. Cyclically loaded components would benefit from smoother surfaces as observed in the case of CMT-welding. Next, the layer volume increases with increasing energy input, which is beneficial for economical manufacturing times.

CT scans did not show severe inner defects or irregularities in the manufactured steel bars, except a low number of single pores with a diameter of ~1000 μm. The differences in pore size and magnitude are explained by the welding time and weld pool size. However, this must be examined statistically in future research with higher numbers of specimens.

The applied test methodology captured both local elastic component behavior and integral material parameters. The full field strain measurements can identify the inhomogeneous material behavior. Most likely, the heterogeneous strain distribution results from geometric variations, i.e., the change of diameter and local notches at layer interfaces. However, future work will address the material properties of deposited weld metal depending on the applied weld process in more detail. The detected stress concentration is of large importance for plasticity-induced phenomena, such as crack initiation and fatigue.

Various parameter identification methods could be used to quantify the deviations of the material in further research, for example the virtual fields method [43]. The measurements on the material scale presented here are not sufficient to evaluate the behavior of more complex WAAM-components in its entirety and 360° digital image correlation systems could be utilized for the purpose of analyzing the deformation behavior of the entire component during a test, even in regions of large deformations where the ESPI system reaches its limits. When analyzing the component behavior, it is also necessary to evaluate the behavior under compressive loads in regards to buckling as the WAAM components are generally slender components.

## 5. Conclusions

This study investigated the use of WAAM steel components in construction. The fundamentals of both architectural and structural design were introduced. The experimental investigations made use of steel bars as the geometrically simplest geometry to demonstrate component and material behavior typical for WAAM structures. The WAAM process was linked to the build-up geometry, surface topography, and material properties. The key findings are:

- The layer geometry depends on the weld energy input. High energy leads to heterogeneous surface topography, higher material volumes deposited, and sharp notches at layer interfaces.
- The heat input (weld energy and interpass temperature) correspond with the microstructure. High heat input results in a homogenous microstructure, but overall lower hardness. Low heat input results in hardness peaks at the layer interfaces.
- CT scans revealed internal and external weld irregularities. Porosity depends on weld pool degassing.
- Heterogeneous surface topography and geometry result in heterogeneous strain distributions under tensile loading.
- Integral material properties depend on the weld process. High heat input leads to low hardness, coarse grains, and lower elongation.

**Author Contributions:** Conceptualization, J.H. and J.U.; methodology, J.H., J.U., and J.M.; investigation, J.M. and M.G.; resources, K.D. and K.T.; writing—original draft preparation, J.M., M.G., and C.M.; writing—review and editing, J.H.; visualization, J.M., M.G., and C.M.; supervision, K.D., K.T., and H.K.; project administration, J.H., K.T., and H.K.

**Funding:** This research received partial funding from Deutsche Forschungsgemeinschaft, Major Research Instrumentation, Optical Strain Measurement System, No. 221277291.

**Acknowledgments:** We acknowledge support by the Open Access Publication Funds of the Technische Universität Braunschweig.

**Conflicts of Interest:** The authors declare no conflict of interest.

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
