# Peer review of "Design and Parameter Identification of Wire and Arc Additively Manufactured (WAAM) Steel Bars for Use in Construction"

_metals, doi:10.3390/met9070725_

Round 1

Reviewer 1 Report

The topic of the manuscript is very interesting and adequate to the journal.

The research was made carefully taking into account many aspects, as microstructure, internal porosity, surface strain distributions, mechanical behaviour, …

Authors also are conscious of the limitations of this research and suggest several tasks to do in future in order to make results more robust.

In my opinion some minor changes must be made before reach a publishable form.

General comments:

Authors divide the experiment in several “phases” (Lines 243, 250, 264 and 388) I think that the term “stages” is more adequate, so please replace “phase” by “stage”.

Use MPa and GPa instead of N/mm2 and strains (in 1/1, i.e. typical materials scientist strains, instead in %0). This is more evident in table 3.

More specific comments and typos:

Line 225. Indicate the manufacturer and model of the Speckle setup used. It looks like a Dantec Dynamics, but specify the brand and model used.

Lines 229: I will replace the first sentence by: “ESPI employs coherent laser that illuminates the sample from different positions and monitor the change in the intensity of the produced interference due to the displacements with a CCD camera. The setup used…”

Line 234: Insert “displacements or strains” between “measurements” and “due”.

Line 309: There is a missed “)” symbol

Table 3: Last row… use the “1” as subindex or superindex, as it looks that correspond to values and not to the footnote (perhaps you can use a “*” instead the “1”.

Lines 404 and 406. Use italic style for ReH, Rm and Rm as you used italic style in the table 3.

Line 416: Use “material properties” instead of “materials parameters”.

p { margin-bottom: 6.25px; direction: ltr; line-height: 115%; text-align: left; background: transparent none repeat scroll 0% 0%; }p.western { }a:link { color: rgb(0, 0, 128); text-decoration: underline; }a:visited { color: rgb(128, 0, 0); text-decoration: underline; }

Author Response

The topic of the manuscript is very interesting and adequate to the journal.

The research was made carefully taking into account many aspects, as microstructure, internal porosity, surface strain distributions, mechanical behaviour, …

Authors also are conscious of the limitations of this research and suggest several tasks to do in future in order to make results more robust.

In my opinion some minor changes must be made before reach a publishable form.

Thank you very much for your in-depth review of this article. We re-worked the article according to your comments.

General comments:

Authors divide the experiment in several “phases” (Lines 243, 250, 264 and 388) I think that the term “stages” is more adequate, so please replace “phase” by “stage”.

Thanks for this Suggestion. We changed it.

Use MPa and GPa instead of N/mm2 and strains (in 1/1, i.e. typical materials scientist strains, instead in %0). This is more evident in table 3.

We changed N/mm² into MPa and for the modulus of elasticty we used GPa. In our oppinion the unit for the ultimate Elongation should be %, since it is a technical Elongation. In the diagram we changed it from 0/00 to %.

More specific comments and typos:

Line 225. Indicate the manufacturer and model of the Speckle setup used. It looks like a Dantec Dynamics, but specify the brand and model used.

Yes, it is a Dantec, Now we specified all equipment used for the experimental setup for tensile testing.

Lines 229: I will replace the first sentence by: “ESPI employs coherent laser that illuminates the sample from different positions and monitor the change in the intensity of the produced interference due to the displacements with a CCD camera. The setup used…”

Thanks for the correction.

Line 234: Insert “displacements or strains” between “measurements” and “due”.

Done. Now ist in line 253

Line 309: There is a missed “)” symbol

Right, Thanks!

Table 3: Last row… use the “1” as subindex or superindex, as it looks that correspond to values and not to the footnote (perhaps you can use a “*” instead the “1”.

Now we changed it to a superindex.

Lines 404 and 406. Use italic style for ReH, Rm and Rm as you used italic style in the table 3.

Alright

Line 416: Use “material properties” instead of “materials parameters”.

Good Point. We also changed it elsewhere.

Reviewer 2 Report

Very interesting paper on WAAM of steel. There are several points that should be addressed as described below. The paper has a lot of potential and the results are of quality.

Some recent critical review works on WAAM could be used to further enrich the state of the art regarding the technology. See for example “Strategies and processes for high quality wire arc additive manufacturing”, “Current Status and Perspectives on Wire and Arc Additive Manufacturing (WAAM)”, and “A review of the wire arc additive manufacturing of metals: Properties, defects and quality improvement”.

Regarding the test surface features in WAAM parts: they should be machined after building up otherwise if used in dynamic loading solidification those surface irregularities will be source for premature failure. This should be revised in the introduction.

Also the thermal history during WAAM greatly impact the properties of the parts, which may not be the same (and often are not) through the part. See for example: “Wire and arc additive manufacturing of HSLA steel: Effect of Thermal Cycles on Microstructure and Mechanical Properties” and “Wire-arc additive manufacturing of a duplex stainless steel: thermal cycle analysis and microstructure characterization” to discuss these important feature.

In figure 3: can the author provide a close up picture of the sample (to see the dimension of the surface irregularities)?

In equation 1 please describe each letter.

From fig 7 it looks like the diameter is higher in c) than in b). However, from fig 6 the results are in opposition to this. Why?

For fig 8 please improve the scale (it’s hard to read) and identify the phases that exist in each macrograph please.

For table 3 specify what is E, Reh, Rm and A.

How many samples were tested until fracture? Only one?

Can the authors provide fracture surface images after tensile testing? This is interesting to see the fracture more and how the microstructure influence these fracture of the specimen.

Author Response

Very interesting paper on WAAM of steel. There are several points that should be addressed as described below. The paper has a lot of potential and the results are of quality.

Thank you very much for your in-depth review of this article. We re-worked the article according to your comments.

Some recent critical review works on WAAM could be used to further enrich the state of the art regarding the technology. See for example “Strategies and processes for high quality wire arc additive manufacturing”, “Current Status and Perspectives on Wire and Arc Additive Manufacturing (WAAM)”, and “A review of the wire arc additive manufacturing of metals: Properties, defects and quality improvement”.

The additional references are discussed and added in lines 91-96.

Regarding the test surface features in WAAM parts: they should be machined after building up otherwise if used in dynamic loading solidification those surface irregularities will be source for premature failure. This should be revised in the introduction.

Discussion of surface topography effects is added to the introduction in lines 166-170.

Also the thermal history during WAAM greatly impact the properties of the parts, which may not be the same (and often are not) through the part. See for example: “Wire and arc additive manufacturing of HSLA steel: Effect of Thermal Cycles on Microstructure and Mechanical Properties” and “Wire-arc additive manufacturing of a duplex stainless steel: thermal cycle analysis and microstructure characterization” to discuss these important feature.

This was added in lines 96-101.

In figure 3: can the author provide a close up picture of the sample (to see the dimension of the surface irregularities)?

Topography scans were additionaly examined by laser scanning. Results are added to Fig. 7

In equation 1 please describe each letter.

The equation is defined and described in l 297.

From fig 7 it looks like the diameter is higher in c) than in b). However, from fig 6 the results are in opposition to this. Why?

Very well spotted. The data evaluation was indeed incorrect. The corrected geometry data is shown in Fig. 6 and Fig.8 respectively. The diameter of Cycle Step is slightly larger than of CMT Standard. Furthermore, it should become now clear that the volume deposited correlates with the energy input (Fig.6).

For fig 8 please improve the scale (it’s hard to read) and identify the phases that exist in each macrograph please.

Fig. 9 now contains clear scales. The microstructure and phase composition is described in 338ff.

For table 3 specify what is E, Reh, Rm and A.

The abbreviations of material properties are added to Tab.3.

How many samples were tested until fracture? Only one?

For this study, several samples were manufactured with each process to identify the influence of welding parameters. However, only one sample of each process was tested until fracture. The overall objective was the identification of an advanced test routine, which can be applied to further WAAM components as well. Further research will cover also statistical evaluation of surface topography, microstructure and mechanical properties.

Can the authors provide fracture surface images after tensile testing? This is interesting to see the fracture more and how the microstructure influence these fracture of the specimen.

SEM images and analysis is added, lines 436ff.

Round 2

Reviewer 2 Report

The manuscript was greatly improved.

Author Response

The manuscript was greatly improved.

Thanks a lot for your review.